# Mei-Gin Formula Ameliorates Obesity through Lipolysis, Fatty Oxidation, and Thermogenesis in High-Fat Diet-Induced Obese Rats

**DOI:** 10.3390/foods12193539

**Published:** 2023-09-22

**Authors:** Hsin-Lin Cheng, Wei-Tang Chang, Jiun-Ling Lin, Chun-Tse Tsai, Ming-Ching Cheng, Shih-Chien Huang, Yue-Ching Wong, Chin-Lin Hsu

**Affiliations:** 1Department of Nutrition, Chung Shan Medical University, Taichung 40201, Taiwan; iamsamlee@livemail.tw (H.-L.C.); jiunling.1209@gmail.com (J.-L.L.); jim888222@gmail.com (C.-T.T.); schuang@csmu.edu.tw (S.-C.H.); wyc@csmu.edu.tw (Y.-C.W.); 2Department of Nutrition and Health Sciences, Chinese Culture University, Taipei 11114, Taiwan; zwt6@ulive.pccu.edu.tw; 3Department of Nutrition, Chung Shan Medical University Hospital, Taichung 40201, Taiwan; 4Department of Health Food, Chung Chou University of Science and Technology, Changhua 51591, Taiwan; m25522@yahoo.com.tw

**Keywords:** Mei-Gin formula, adipose tissue, anti-obesity effect, high-fat diet-induced obesity

## Abstract

Obesity is a metabolic dysfunction characterized by excessive body fat deposition as a consequence of an energy imbalance. Novel therapeutic strategies have emerged that are safe and have comparatively low side effects for obesity treatment. Functional foods and nutraceuticals have recently received a great deal of attention because of their components with the properties of antimetabolic syndrome. Based on our previous in vitro and in vivo investigations on anti-adipogenesis activity and improved body fat accumulation in serials, the combination of three ingredients (including bainiku-ekisu, black garlic, and *Mesona procumbens* Hemsl), comprising the Mei-Gin formula (MGF), was eventually selected as a novel inhibitor that exhibited preventive effects against obesity. Herein, we verify the anti-obesity effects of MGF in obese rats induced by a high-fat diet and discuss the potential molecular mechanisms underlying obesity development. Oral administration of MGF significantly suppressed the final body weight, weight change, energy and water intake, subcutaneous and visceral fat mass, liver weight, hepatic total lipids and triglycerides (TG), and serum levels of TG, triglycerides (TC), low-density lipoprotein cholesterol (LDL-C), alanine transaminase (AST), uric acid, and ketone bodies and augmented fecal total lipids, TG, and cholesterol excretion in the high-dose MGF-supplemented groups. Furthermore, the corresponding lipid metabolic pathways revealed that MGF supplementation effectively increased lipolysis and fatty acid oxidation gene expression and attenuated fatty acid synthesis gene expression in the white adipose tissue (WAT) and liver and it also increased mitochondrial activation and thermogenic gene expression in the brown adipose tissue (BAT) of rats with obesity induced by a high-fat diet (HFD). These results demonstrate that the intake of MGF can be beneficial for the suppression of HFD-induced obesity in rats through the lipolysis, fatty oxidation, and thermogenesis pathway. In conclusion, these results demonstrate the anti-obesity efficacy of MGF in vivo and suggest that MGF may act as a potential therapeutic agent against obesity.

## 1. Introduction

The incidence of obesity has been steadily increasing for several decades and become a global public health and economic problem [1]. The condition of obesity is characterized by abnormal or excessive body fat accumulation, resulting in adipose tissue expansion [2,3]. It is a risk factor contributing to serious chronic complications, including hyperglycemia, cardiovascular disease, hypertension, and malignant diseases [4]. WAT primarily acts as an energy reservoir to store excess calories as TG in lipid droplets [5]. Pharmacological strategies for prevention and treatment mainly rely on the interruption of pathological WAT accumulation and involve the reduction of adipogenesis and improving mitochondrial β-oxidation via lipolysis [6,7,8,9]. BAT is postulated to dissipate energy and act as a target against the development of obesity by improving the thermogenesis process, and it thus appears to play a crucial role in the anti-obesity strategy [10,11,12,13]. Currently, emerging studies highlight the therapeutic potential of natural products with potent weight loss properties that might serve as anti-obesity agents. 

Traditionally, edible herbs have served as folk medicines that are widely used in Taiwan to treat chronic liver injury, inflammation, heat shock, hypertension, and hyperglycemia [14,15,16,17]. Previously, the antibacterial activity of bainiku-ekisu (*Prunus mume* juice concentrate) was reported against Helicobacter pylori strains by conducting an in vivo and in vitro pilot study [18,19]. Intriguingly, Yang et al. previously reported the positive effects of bainiku-ekisu on metabolic disorders, which could be attributed to the fact that the total phenolic content was higher than that of fresh apricot juice due to the concentration process [20]. The herb *Mesona procumbens* Hemsl. (Hsian-tsao) is a natural herbal drink and jelly-like dessert in the Orient. Aqueous extracts of *Mesona procumbens* Hemsl were established to exert myocardium protection activity in an STZ-induced rat model [21]. Phenolic compounds in *Mesona procumbens* Hemsl extracts may act as free radical scavengers and have been documented to demonstrate possible protective actions [22]. Black garlic is a processed product generated from raw garlic (*Allium sativum* L.). Previously, the beneficial effects of black garlic were reported to be associated with changes in nutrient composition during fermentation, which were attributed to the higher phytochemical components in black garlic than in fresh garlic. Extensive studies showed that black garlic might improve blood lipid profiles; in a randomized controlled study [23], it inhibited NO and TNF-α production dose-dependently through the intracellular MAPK signal pathway in endotoxin-induced RAW 264.7 macrophages [24]. Therefore, functional foods and nutraceuticals have attracted much attention as an available therapeutic approach for the prevention and treatment of obesity. 

According to preliminary investigations, we previously developed an innovative herbal formula consisting of three ingredients (bainiku-ekisu, black garlic, and *Mesona procumbens* Hemsl), called the Mei-Gin formula (MGF), which also demonstrated its potent anti-obesity effects in vivo and in vitro. Interestingly, each component of the MGF has been proven to have strong biological activity relevant to disease treatment, as described in the previous paragraph. Among the Mei-Gin formulas (MGF−1-7), we found that MGF−7 exhibited the optimal anti-obesity efficacy by suppressing body weight, liver weight, and total body fat in a HFD-induced obese rat model [25]. Functional substance content analysis was conducted using high-performance liquid chromatography (HPLC), which revealed high phenolic acid content, including *p*-coumaric acid, caffeic acid, and chlorogenic acid. In recent years, numerous studies have revealed the synergistic effects of herbal formula extracts that provide significant and powerful therapeutic activity [25,26,27,28]. Based on the above facts, the beneficial effects of this innovative formula against obesity deserve further investigation. This present study evaluates the efficacy of MGF in the HFD-induced rat model and aims at verifying the therapeutic dosage with a low and/or safer concentration. Promising results showed the anti-obesity effects of MGF, and the underlying molecular mechanism involved in the regulation was further clarified. 

## 2. Materials and Methods

### 2.1. Chemicals and Reagents

The cholesterol assay kit was purchased from Randox Laboratories Co., Ltd. (Antrim, UK). The E.Z.N.A™ tissue RNA kit was from Omega Bio-Tek (Norcross, GA, USA). The high-capacity RNA-to-cDNA kit and the power SYBR^®^ Green RT-PCR reagent kit were obtained from Applied Biosystems (Richmond, CA, USA). The TG-GPO liquid reagent set was obtained from Teco Diagnostics (Anaheim, CA, USA).

### 2.2. Preparation of Mei-Gin Formula−7

The MGF−7 was kindly offered by Professor Ming-Ching Cheng, and the content of the Mei-Gin formula was as described previously. Briefly, the ingredients of the Mei-Gin formula were obtained from concentrated *Prunus mume* juice (Bainiku-ekisu), the ethanol extracts of *Mesona procumbens* Hemsl. and *Prunus mume* fruit, and the aqueous extract of black garlic acid. The HPLC method was conducted to identify the phenolic content of MGF−7, including chlorogenic acid (0.72 mg/g), caffeic acid (0.11 mg/g), and p-coumaric acid (0.08 mg/g).

### 2.3. Design of Animal Study

A total of sixty male Wistar rats were obtained from BioLASCO Taiwan Co., Ltd. (Taipei, Taiwan). The rats were housed in cages under a controlled-atmosphere environment (23 ± 2 °C at 60% relative humidity) with 12:12 light–dark cycles at the Chung Shan Medical University Animal Facility. Each intervention dose of MFD−7 was determined according to guidance from the Food and Drug Administration (FDA) at the initial dose that was equal to a 60 kg healthy adult. Equivalent dose conversion from human to rat was based on the body surface and calculated according to the following equation: human dose in mg/kg = animal dose in mg/kg (human km = 37 and rat km = 6). The rats were fed a laboratory chow diet for 7 days and then randomly allocated into the following groups (*n* = 12/ group): (i) the ND group, fed an AIN-93G control diet (7% fat); (ii) the HFD group, fed a high-fat diet containing 32% lipids; (iii) the HFD + MGF7−LD group, fed HFD with 50 mg/kg/day MGF−7, 1X; (iv) the HFD + MGF7−MD group, fed HFD with 100 mg/kg/day MGF−7, 2X; (v) the HFD + MGF7−HD group, fed HFD with 300 mg/kg/day MGF−7, 6X. The diet formulations are described in detail in Table 1. The rat body weight, feed intake, and water intake were measured during the diet period and used to calculate the feed efficiency. Three days before the end of the intervention, a fecal sample was collected and lyophilized. At the end of the experiment, the rats were fasted overnight before euthanasia by asphyxiation of CO_2_. Blood samples, organs (heart, liver, spleen, lung, and kidney), and adipose tissue (perirenal, epididymal, mesenteric, retroperitoneal, inguinal, and brown adipose fat) were collected, weighed, and stored at −80 °C until analyzed. The experimental procedures were registered at the following website: https://preclinicaltrials.eu (accessed on 14 September 2023) (PCT ID: PCTE0000404).

### 2.4. Biochemical Analysis 

Blood was taken 8 weeks after the intervention period and placed in a plastic tube (BD Vacutainer, Plymouth, UK). Serum was obtained after 15 min centrifugation at 4000× *g*. The serum glucose level was determined using a commercial assay ELISA kit (Teco Diagnostics, Randox Laboratories Co., Ltd., Antrim, UK). Serum total cholesterol (TC), low-density lipoprotein cholesterol (LDL-C), and high-density lipoprotein cholesterol (HDL-C) were measured using a commercial assay ELISA kit (Denka Seiken Co., Ltd., Tokyo, Japan). The AST, ALT, urea acid, creatinine, Na^+^, K^+^, and Cl^–^ levels in serum were analyzed using a commercial assay ELISA kit (DiaSys Co., Ltd., Holzheim, Germany). Serum ketone body and TG concentrations were determined using a commercial assay ELISA kit assay (Randox Laboratories Co., Ltd., Antrim, UK). 

### 2.5. Lipid Content Analysis of Liver and Feces 

The liver tissue and feces were subjected to lipid content extraction and the procedure was performed as described elsewhere [29]. The concentrations of TG and TC in the liver and feces were measured using a TG assay kit (Teco Diagnostics, Anaheim, CA, USA) and TC assay ELISA kit (Cholesterol, Randox Laboratories Co., Ltd., Antrim, UK), respectively.

### 2.6. Histological Analysis

The liver and perirenal adipose tissue were fixed in 10% formalin-buffered solution and embedded in paraffin. Tissue sections were stained with hematoxylin and eosin (H&E) using standard techniques. The sections were photographed by light microscopy (AxioCam ERc 5s, Carl Zeiss Microscopy GmbH, Jena, Germany) and the size of the adipocytes was determined using the Image J software version 1.53k. 

### 2.7. RNA Extraction and Quantitative Real-Time Polymerase Chain Reaction (qRT-PCR)

The levels of obesity-related gene expression in the liver and adipose tissue were examined by qRT-PCR analysis. Total RNA was isolated from the liver, perirenal adipose, and brown adipose tissue using the E.Z.N.A™ tissue RNA kit (Omega Bio-Tek, USA), according to the manufacturer’s instructions. Total RNA was used to synthesize cDNA using the ABI high-capacity cDNA reverse transcription kit (Applied Biosystems, Foster City, CA, USA). The specific primers used for the amplification of the analyzed genes were as follows, including forward (F) and reverse (R) primer sequences: *ACO*-F: 5′-CACGCAATAGTTCTGGCTCA-3′, *ACO*-R: 5′-ACCTGGGCGTATTTCATCAG-3′ (GI: 666184579); *Adiponectin*-F: 5′-CTCCACCCAAGGAAACTTGT-3′, *Adiponectin*-R: 5′-CTGGTCCACATTTTTTTCCT-3′ (GI: 46485455); *AMPK*-F: 5′-ACACCTCAGCGCTCCTGTTC-3′, *AMPK*-R: 5′-CTGTGCTGGAATCGACACT-3′ (GI: 109657628); *ATGL*-F: 5′-TGTGGCCTCATTCCTCCTAC-3′, *ATGL*-R: 5′-AGCCCTGTTTGCACATCTCT-3′ (GI: 189095276); *β*-*actin*-F: 5′-TACAATGAGCTGCGTGTGG-3′, *β*-*actin*-R: 5′-TGGTGGTGAAGCTGTAGCC-3′ (GI: 55574); *CPT*-*1*-F: 5′-GCTCGCACATTACAAGGACAT-3′, *CPT*-*1*-R: 5′-TGGACACCACATAGAGGCAG-3′ (GI: 255652919); *FAS*-F: 5′-CTTGGGTGCCGATTACAACC-3′, *FAS*-R: 5′-GCCCTCCCGTACACTCACTC-3′ (GI: 204098); *FATP1*-F: 5′-GTGCGACAGATTGGCGAGTT-3′, *FATP1*-R: 5′-GCGTGAGGATACGGCTGTTG-3′ (GI: 568815579); *FOXO1*-F: 5′-GCTGGGTGTCAGGCTAAGAG-3′, *FOXO1*-R: 5′-GCATCTTTGGACTGCTCCTC-3′ (GI: 393794775); *HSL*-F: 5′-CCCATAAGACCCCATTGCCTG-3′, *HSL*-R: 5′-CTGCCTCAGACACACTCCTG-3′ (GI: 109644432); *Perilipin*-F: 5′-AGCGAGGATGGCAGTCAAC-3′, *Perilipin*-R: 5′-GATGCTGTTTCTGGCACTG-3′ (GI: 101795741); *PGC*-*1β*-F: 5′-TTGACAGTGGAGCTTTGTGG-3′, *PGC*-*1β*-R: 5′-GGGCTTATATGGAGGTGTGG-3′ (GI: 31341573); *PPAR*-*α*-F: 5′-CATCGAGTGTCGAATATGTGG-3′, *PPAR*-*α*-R: 5′-GCAGTACTGGCATTTGTTCC-3′ (GI: 238018083); *SIRT1*-F: 5′-GATCTCCCAGATCCTCAAGCC-3′, *SIRT1*-R: 5′-CACCGAGGAACTACCTGAT-3′ (GI: 157822758); *UCP*-*1*-F: 5′-ACACTGTGGAAAGGGACGAC-3′, *UCP*-*1*-R: 5′-CATCTGCCAGTATGTGGTGG-3′ (GI: 666183369). PCR amplification was performed in the mixture of a specific primer and PowerUp SYBR Green Master Mix (Applied Biosystems, Foster City, CA, USA) with the Applied Biosystems real-time PCR system (StepOne, Applied Biosystems, Foster City, CA, USA). The relative expression levels of all the genes were normalized to β-actin.

### 2.8. Statistical Analysis

All of the measurement results are presented as the mean ± SEM. The data analyses of variance were performed by one-way ANOVA with Duncan’s multiple range tests. The statistical differences were considered to be significant when the *p* values were below 0.05 (IBM SPSS 22.0, Chicago, IL, USA). 

## 3. Results

### 3.1. Effect of the Mei-Gin Formula on Body Weight, Food Utilization, and Organ Weights in HFD-Induced Obese Rats

To explore the in vivo anti-obesity effect of the Mei-Gin formula, the rats were fed with HFD or HFD supplemented with MGF for eight weeks. As shown in Figure 1, the HFD + MGF groups (LD, MD, and HD) showed attenuated body weight gain compared to the HFD group during the dietary intervention. Furthermore, the final body weights in the HFD + MGF groups (LD, MD, and HD) were significantly lower than in the HFD group. No significant differences were observed in feed intake, energy intake, and water intake between the HFD group and the HFD + MGF groups (LD, MD, and HD). A significant decrease in feed efficiency was observed in the HFD + MGF groups (LD, MD, and HD) as compared with the HFD group. Similarly, a significant reversal was found in the increased weight of the liver induced by HFD supplementation. However, the weights of the heart, spleen, lung, and kidney showed no significant differences between each group (Table 2). 

### 3.2. Effect of the Mei-Gin Formula on Fat Accumulation in HFD-Induced Obese Rats

As shown in Table 3, HFD supplementation significantly increased the total body fat, visceral adipose tissue, and subcutaneous adipose tissue mass in the HFD group compared to the ND group. Compared to the HFD group, the HFD + MGF groups (LD, MD, and HD) showed a significant decrease in the HFD-induced alterations in total body fat, visceral adipose tissue, and subcutaneous adipose tissue mass. Interestingly, although no significant differences were found between the HFD + MGF group (LD) and the HFD group, it still exhibited potent activity in reducing visceral adipose mass. Among visceral adipose tissue, the HFD + MGF group (HD) showed a significant reduction in the weight of the perirenal adipose tissue. Furthermore, the HFD-induced increase in inguinal adipose tissue mass was reversed by MGF supplementation. No significant differences were observed in brown adipose tissue between each group. 

### 3.3. Effect of the Mei-Gin Formula on Serum Lipid Profiles in HFD-Induced Obese Rats

As shown in Table 4, HFD supplementation produced a significant decrease in serum TG and TC levels compared to the ND group. When compared with the HFD group, the HFD + MGF group (HD) showed a significant decrease in the serum level of TG, as well as in the TC level in the HFD + MGF groups (MD and HD). No significant differences were observed in serum high-density lipoprotein cholesterol (HDL-C), low-density lipoprotein cholesterol (LDL-C), LDL-C/HDL-C ratio, aspartate transaminase (AST), alanine transaminase (ALT), Na^+^, K^+^, and Cl^–^ between the ND group and the HFD group. A lower serum level of HDL-C, ALT, uric acid, and ketone body was observed after MGF supplementation, while the serum level of LDL-C was found to decrease in the HFD + MGF group (MD and HD). Moreover, the LDL-C/ HDL-C ratio was significantly lower in the HFD + MGF group (HD) compared to the HFD group. A lower uric acid level was also observed in the HFD + MGF group (LD) than the HFD group. 

### 3.4. Effect of Mei-Gin Formula on Hepatic Lipid Content and Adiposity in HFD-Induced Obese Rats

A histological examination was performed to explore the effects of MGF on lipid metabolism. The histological analysis of rats’ H&E-stained liver and perirenal adipose tissue is shown in Figure 2. Severe hepatosteatosis with the extensive vacuolization and accumulation of lipid droplets was observed in HFD supplementation rats, while rats in the ND group exhibited lined hepatic sinusoids without the infiltration of lipid droplets or hepatocyte swelling. In contrast, MGF-supplemented rats exhibited reduced lipid droplet accumulation in hepatic intracellular vacuoles. The size of the perirenal adipose tissue cells presented a distended arrangement in rats fed the HFD, while the supplementation of MGF significantly reversed the HFD-induced increase in cell size in this area.

### 3.5. Effect of Mei-Gin Formula on the Hepatic and Fecal Lipid Profiles in HFD-Induced Obese Rats 

The effect of HFD supplemented with MGF on the hepatic and fecal total lipids, TG, and cholesterol of rats is shown in Figure 3. Supplementation with HFD produced considerably higher levels of total lipids, TG, and cholesterol levels than in the ND group. The HFD + MGF group (HD) exhibited the lowest levels of total liver lipids and TG, while the hepatic TG and cholesterol levels in the HFD + MGF groups (MD and HD) were significantly decreased. However, hepatic cholesterol in the MGF-supplemented groups did not differ significantly from that in the HFD group. Furthermore, the results also showed that MGF supplementation led to an increase in fecal lipids, TG, and cholesterol secretion in the HFD + MGF groups (MD and HD). 

### 3.6. Effect of Mei-Gin Formula on Gene Expression in Hepatic Lipid Metabolism and WAT of HFD-Induced Obese Rats

To further explore the reduction in hepatic steatosis and fat mass in MGF-supplemented rats and whether it is accompanied by altered lipid metabolism, quantitative RT-PCR was performed. Compared with the HFD group, the liver β-oxidation-related gene expression of *AMPK* (MD and HD), *PGC*-*1β* (HD), *PPAR*-*α* (MD and HD), *CPT*-*1* (HD), and *ACO* (MD and HD) was significantly increased after MGF supplementation (Figure 4A). Furthermore, the genes associated with fatty acid synthesis, *SIRT1* and *FOXO1*, were increased in the HFD + MGF group (HD) as compared to the HFD group (Figure 4B). In the HFD + MGF group (HD), the gene expression of *ATGL* and *HSL* significantly increased, which indicated an enhancement in lipolysis (Figure 4C). Furthermore, a similar tendency for gene expression involved in lipid metabolism was found in the perirenal adipose tissue. An elevated trend was observed in the mRNA levels of *PGC*-*1β* (MD and HD), *CPT*-*1* (LD, MD, and HD), *ACO* (MD and HD), and *UCP*-*1* (MD and HD) after MGF supplementation (Figure 4D). Furthermore, the expression of the *SIRT1* (LD, MD, and HD) and *FOXO1* (MD and HD) genes was elevated after MGF supplementation, while the level of *FAS* (MD and HD) was significantly decreased compared with that of the HFD group (Figure 4E). MGF significantly enhanced lipolysis by increasing the expression of *ATGL* (LD, MD, and HD) and *HSL* (MD and HD) mRNA and decreasing the expression of *perilipin* (MD and HD) (Figure 4F). Furthermore, we also observed that the HFD + MGF group (HD) had significantly lower expression of *FATP* mRNA (Figure 4G). In contrast, the HFD + MGF groups (MD and HD) had markedly increased adiponectin mRNA expression compared with the HFD group (Figure 4H). 

### 3.7. Effect of the Mei-Gin Formula on Thermogenesis and Mitochondrial Biogenesis and Activity in BAT of HFD-Induced Obese Rats

In addition to the direct effects of altering lipid metabolism via supplementation with MGF in HFD-induced obese rats, we further investigated the regulation of gene expression, which was also mediated by active energy expenditure to protect against obesity in BAT through thermogenesis and mitochondrial activity. Regulatory factors, including *AMPK* (MD and HD) *BMP*-*7* (MD and HD), and *PGC*-*1α* (HD), were significantly increased, while the expression of the *ACC* gene (LD, MD, and HD) was significantly decreased in the BAT of the MGF-supplemented groups (Figure 5A). Furthermore, the thermogenic gene expression of *C/EBPβ* (MD and HD), *PPAR*-*γ* (HD), *PRDM16* (MD and HD), *UCP*-*1* (HD), and *Ebf2* (HD) was significantly increased after MGF supplementation (Figure 5B). 

## 4. Discussion

Obesity is a cluster of chronic metabolic diseases, characterized by excessive body fat accumulation and elevated levels of lipids in the blood [30,31]. The diet-induced obesity rodent is an important therapeutic model in elucidating obesity-related syndromes including abnormal fat deposition and hyperlipidemia due to the similarity in the pathogenesis of human obesity [32]. As a strategy to treat or prevent obesity, phytochemical compounds present in plant foods need comprehensive investigation and clarification of the underlying mechanisms. At the end of the intervention period, a substantial increase in final body weight, weight gain, and feed efficiency was established in the HFD group as compared to the ND group [33]. Previously, studies have indicated that the consumption of a high-fat diet can cause hyperlipidemia, enlarged fat mass, and lipid deposits in target tissues, such as the liver and adipose tissues, in rodents [34,35]. Meanwhile, a high-fat diet in combination with MGF effectively reduced the final body weight, weight gain, and feeding efficiency compared with the HFD group. Consistently, our results demonstrated that HFD supplemented with MGF considerably restrained weight gain and total body fat (visceral and subcutaneous adipose tissue). In our previous pilot study, dietary supplementation of our novel Mei-Gin formula containing three main hydroxycinnamic acid compounds, including chlorogenic acid, caffeic acid, and *p*-coumaric acid, in obese rats fed a high-fat diet resulted in a remarkable reduction in final body weight and weight gain; moreover, it significantly improved the liver weight and body fat mass (visceral and subcutaneous adipose tissue). Moreover, the histological analysis also exhibited significantly reduced hepatic lipid accumulation and adipocyte sizes in the MGF-supplemented groups, which is consistent with our previous in vitro model of adipogenesis in the 3T3-L1 adipocyte. Consequently, supplementation with dietary phenolic substances improved fecal excretion accompanied by increased calorie, protein, and lipid excretion. Interestingly, MGF supplementation not only increased fecal lipid excretion but also improved hepatic lipid profiles. Elevated serum liver enzymes AST and ALT have been shown to be associated with the presence of hepatic steatosis and fibrosis [36]. Observations of the organ weights and tissue sections of the liver, and the serum AST and ALT levels, confirmed a resulting increase in liver lipid content deposited, which was later improved by MGF supplementation. Furthermore, the serum lipid profiles of TG, TC, and LDL-C levels were also lowered by the MGF supplementation compared to the HFD group. Similar observations were made in other investigations, such as that of Kim and coworkers, who indicated that a cholesterol- and fat-enriched diet increased the serum LDL-C, TG, and TC, hepatic levels of AST and ALT in rodents with cardiometabolic syndrome. Sulfated polysaccharide-rich *Caulerpa racemosa* administration led to a significant reduction in elevated serum lipid profiles, as well as final body weight, particularly in the high-dose-fed group [37,38]. However, our rats fed a high-fat diet did not show increased serum levels of TG, TC, and LDL-C, which could be due to low-carbohydrate dietary effects [39,40]. Lamont et al. compared the effects of a normal chow diet and low-carbohydrate high-fat diet (LCHFD) on NZO mice and the results indicated that mice fed the LCHFD had significantly decreased plasma TG levels, but increased HDL-C levels [40]. Interestingly, typical symptoms of obesity were observed in LCHFD mice, including an elevated body weight and gonadal fat mass, which was consistent with our findings in the HFD group. In the present study, MGF-supplemented rats showed anti-obesity effects through decreased body weight gain and adipose tissue, as well as serum levels of TG, TC, and LDL-C. 

Hydroxycinnamic acid compounds are often seen as phenolic substances present in flowering plants. It has well-documented anti-aging, anti-inflammation, depigmentation, and anti-microbial activity, as well as other health effects [41,42,43]. Hydroxycinnamic acid derivatives obtained from esterification and amidation have also exhibited diverse biomedical and industrial potential. Properties based on the presence of functional hydroxyl groups have led to its wide use as a substance and ingredient in cosmeceutical formulations [43,44]. Hydroxycinnamic acids can also be found in food plants as quinic acid and glucose in the form of simple esters with the following monocaffeoylquinic acids: chlorogenic acid and neochlorogenic acid [45,46]. Furthermore, *p*-coumaric acid, caffeic acid, and ferulic acid are the most common derivative forms of benzoic acid, including *p*-hydroxybenzoic acid, vanillic acid, and protocatechuic acid [47]. Accumulated evidence has showed the synergistic effects of phytochemicals in preventing chronic metabolic disorders and malignant diseases [48,49,50]. Leng and colleagues were the first to combine crocin, chlorogenic acid, geniposide, and quercetin and reveal the synergistic anti-hyperlipidemic effects through the modulation of cholesterol synthesis, hepatic lipid accumulation, and related gene expression [51]. In this respect, another study recently identified a group of phenolic compounds in extracts of red tomato and its products as hydroxycinnamates, which include chlorogenic acid, caffeic acid, ferulic acid, and *p*-coumaric acid, with anti-platelet action [52]. In fact, our preliminary findings confirmed for the first time the anti-obesity potential of MGF regarding weight control and lipid metabolism in vitro and in vivo. In the present study, we demonstrated that the anti-obesity effects of this novel formula appear to be mediated by improved weight gain, body fat mass, lipid excretion, and biochemical patterns in the fatty liver due to abnormal lipid accumulation. In this sense, the functional mechanism of MGF is assumed to be mediated by the regulation of specific markers and pathways involved in fatty acid metabolism. 

Previously, high-fat diets in rats have been reported to induce the gene expression of fatty acid synthesis, lipid transport, and storage and inhibit the expression of adipokines, lipolysis, and β-oxidation genes [53,54]. To assess whether the reduced hepatosteatosis and fat accumulation induced by MGF could be explained by alterations in the aforementioned pathway of lipid metabolism, related gene expression levels were investigated. Since HFD-induced obese animals are a well-established model, we previously demonstrated that rodents fed with a HFD (based on AIN-93G) had significantly reduced gene expression involved in liver fatty acid synthesis and increased gene expression in adipose fatty acid oxidation [33]. Herein, the supplementation of MGF exhibited a robust ability to promote energy expenditure while inhibiting excessive storage by increasing the mRNA expression of *ATGL*, *HSL*, *AMPK*, *PGC*-*1*β, *PPAR*-*α*, *CPT*-*1*, and *ACO* and decreasing *Perilipin* mRNA expression, which is involved in modulating lipolysis and fatty acid oxidation; it also inhibited fatty acid synthesis by decreasing *SIRT1* and *FOXO1* mRNA levels in the liver. This is similar to the result for perirenal adipose tissue, where *ATGL* and *HSL* mRNA levels were significantly higher, which revealed better enzymatic control of the lipolysis pathway. As expected, the rate of metabolic expenditure increased, by which *PGC*-*1β*, *CPT*-*1*, *ACO*, and *UCP*-*1* were expressed more significantly in obese rats with MGF supplementation. Furthermore, the mRNA expression of *SIRT1* and *FOXO1* slightly increased, while *FAS* was subsequently decreased in the perirenal adipose tissue, which demonstrated a reduction in fatty acid synthesis and therefore reduced the development of hyperlipidemia. Adiponectin served as a coordinator of fatty acid oxidation through the activation of AMPK in adipocytes. Similarly to the reduction in fatty acid oxidation, the expression level was increased, whereas, as expected, the *FATP*-*1* level was decreased after MGF supplementation. Although hydroxycinnamic acids are often described as free radical scavengers and strong growth inhibitors of several strains of bacteria, they are now widely recognized as effective anti-adipogenic substances, and they can decrease fat deposition, improve serum lipid profiles, and increase liver lipid metabolism [55,56,57]. Supplementation with chlorogenic acid decreased circulating triacylglycerol and cholesterol levels in Zuker diabetic rats, as well as plasma cholesterol in HFD-induced obese ICR mice. Additionally, chlorogenic acid and caffeic acid decreased fatty acids while increasing fatty acid oxidation by stimulating the expression of PPAR-α in the liver. In an in vitro 3T3-L1 model, *p*-coumaric acid stimulated the secretion of adiponectin and antioxidant enzymes such as superoxide dismutase (SOD), glutathione (GSH), and glutathione S-transferase (GST) in TNF-α-stimulated cell inflammation. In addition to the regulatory pathways of lipid metabolism in the liver and WAT, we further explored whether the capacity for thermogenesis and mitochondrial activity might also contribute to inducing energy expenditure after MGF supplementation. In the MGF-supplied groups, MGF appeared to induce *UCP*-*1* gene expression and increase other genes responsible for mediating the thermogenic effect (*C/EBPβ*, *PPAR*-*γ*, *PRDM16*, and *Ebf2*) in BAT. Furthermore, elevated mitochondrial biogenic genes (*AMPK*, *BMP*-*7*, and *PGC*-*1α*) of BAT also suggested induction in the thermogenic program. Consistent with the reduction in body weight gain, the BAT of MGF-supplied rats appeared to exhibit a higher energy dissipation rate, which improved obesity. In agreement with previous evidence that the weight loss in HFD-induced animal models is induced by the HFD through the activation of BAT and induction of thermogenesis in BAT, which can induce an anti-obesity effect [58,59], the aforementioned data provide direct evidence that hydroxycinnamic acid, especially in our MGF, reduced HFD-induced obesity and the associated complications. However, the present study had both advantages and disadvantages. The main limitation of the study is the lack of biological relevance of the impact of MGF on the serum biochemistry and body composition in normal rats. Although this makes it difficult to determine the basic effects of MGF and detail its actions against HFD-induced obesity, our results provide in vivo evidence of the significant anti-obesity properties of MGF in diet-induced obesity. 

## 5. Conclusions

In conclusion, we report the pre-clinical significant finding that the innovative herbal formula MGF possesses diverse actions targeting obesity pathology cascades in diet-induced obese rats. Our data demonstrate that MGF significantly reduced body weight gain, fat mass, and hepatic lipid profiles, and improved fecal lipid excretion. Furthermore, these in vivo findings indicate that lipolysis, fatty oxidation, and thermogenesis, as central mechanistic pathways generated through MGF, exhibit a strong effect against HFD-induced obesity in rats (Figure 6). This work shows the great synergistic anti-obesity efficacy of combined phenolic substances, which may be used as a natural and safe therapeutic agent for the treatment of obesity.

## Figures and Tables

**Figure 1 foods-12-03539-f001:**
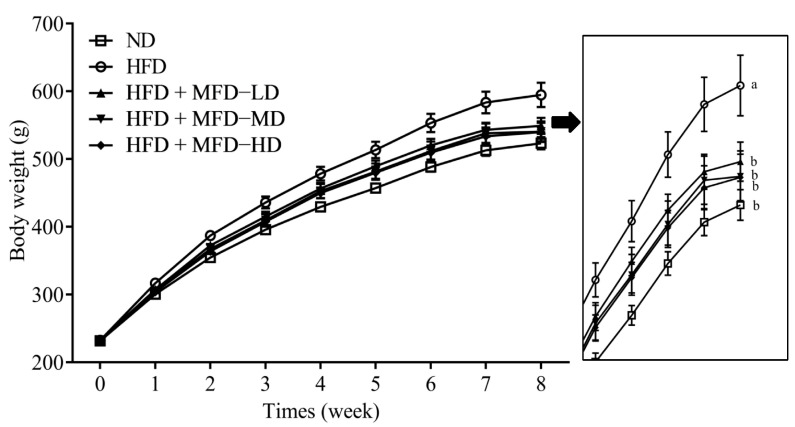
Effect of the Mei-Gin formula on body weight change in high-fat diet-induced obese rats. Data are presented as the mean ± SEM. Different letters for each test parameter are considered as statistical significance (*p* < 0.05). ND, normal diet; HFD, high-fat diet; HFD + MGD−LD, HFD treatment with low-dose Mei-Gin formula (50 mg/kg rat); HFD + MGD−MD, HFD treatment with medium-dose Mei-Gin formula (100 mg/kg rat); HFD + MGD−LD, HFD treatment with high-dose Mei-Gin formula (300 mg/kg rat).

**Figure 2 foods-12-03539-f002:**
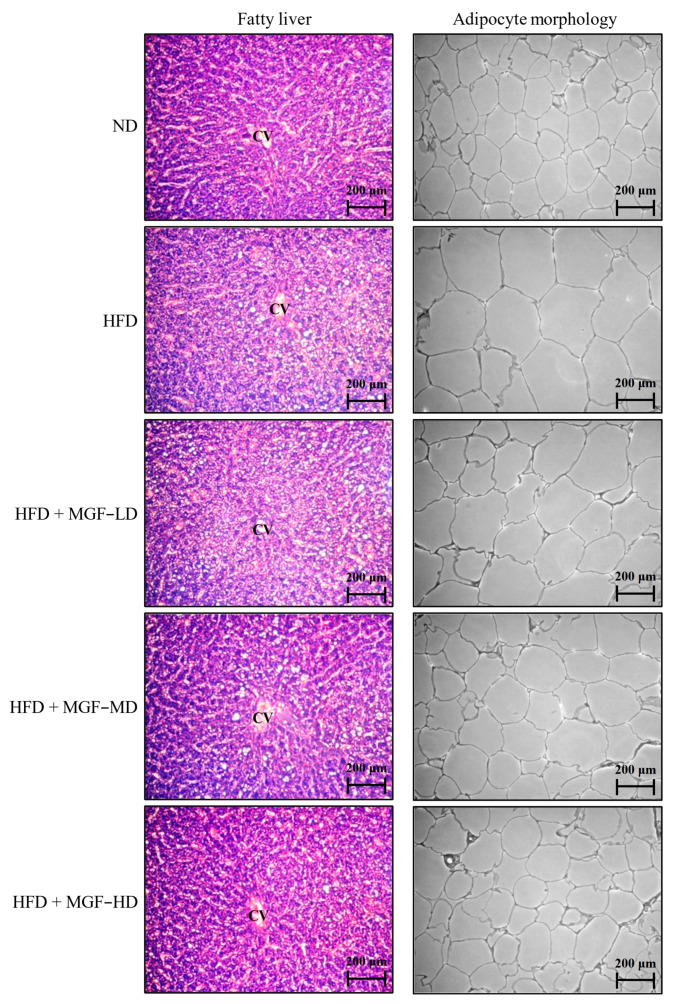
Effect of the Mei-Gin formula on hepatic lipid droplet (**left**) and perirenal adipocyte size (**right**) in obese rats induced by high-fat diet. The liver and perirenal adipose tissue sections were stained with hematoxylin and eosin (H&E). Original magnification: 200×.

**Figure 3 foods-12-03539-f003:**
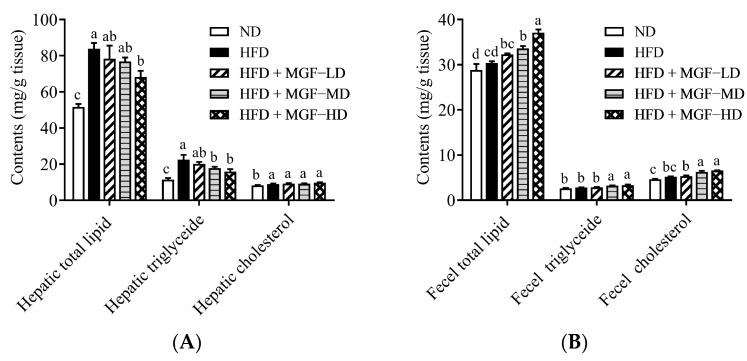
Effect of Mei-Gin formula on hepatic and fecal lipid profiles in obese rats induced by a high-fat diet. (**A**) Hepatic and (**B**) fecel lipid contents in each groups. Data are presented as the mean ± SEM. Different letters for each test parameter are considered for statistical significance (*p* < 0.05).

**Figure 4 foods-12-03539-f004:**
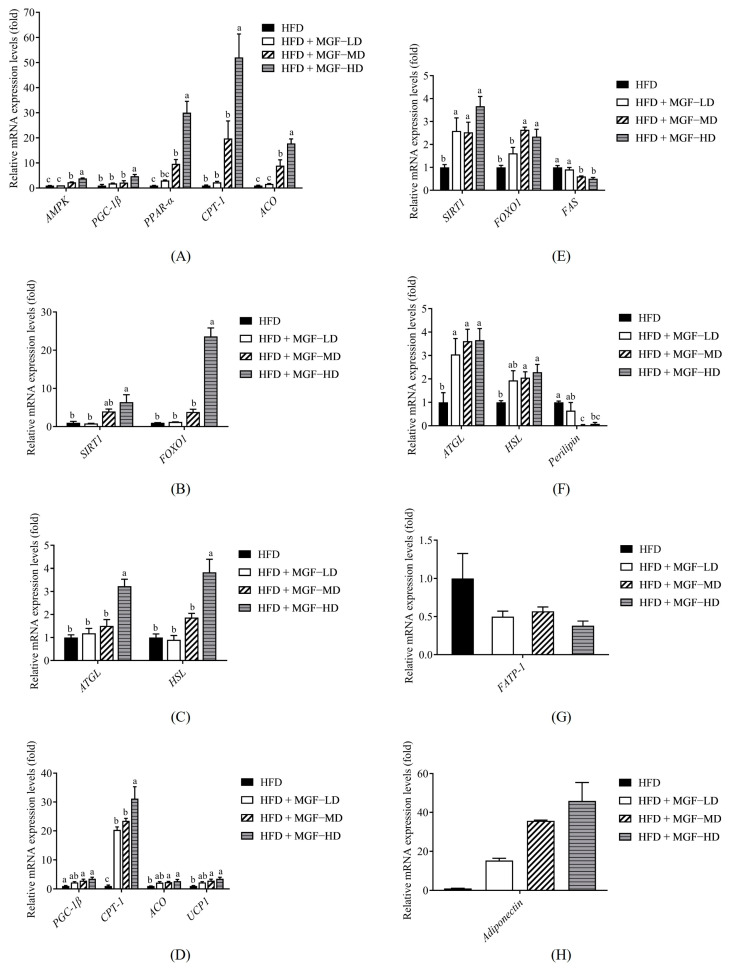
Effect of the Mei-Gin formula on gene expression of β-oxidation, fatty acid synthesis, lipolysis, and lipid transport and storage in liver (**A**–**C**) and perirenal adipose tissue (**D**–**H**) in obese rats induced by a high-fat diet. Data are presented as the mean ± SEM. Different letters for each test parameter are considered for statistical significance (*p* < 0.05). *AMPK*, AMP-activated protein kinase; *PGC-1β*, peroxisome proliferator-activated-gamma coactivator 1 beta; *PPAR-α*, peroxisome proliferator-activated receptor-alpha; *CPT-1*, carnitine palmitoyltransferase 1; *ACO*, acyl-CoA oxidase; *SIRT-1*, sirtuin 1; *FOXO1*, forkhead box protein O1; *ATGL*, adipose TG lipase; *HSL*, hormone-sensitive lipase; *UCP1*, uncoupling protein 1; *FAS*, fatty acid synthase; *FATP-1*, fatty acid transport protein 1.

**Figure 5 foods-12-03539-f005:**
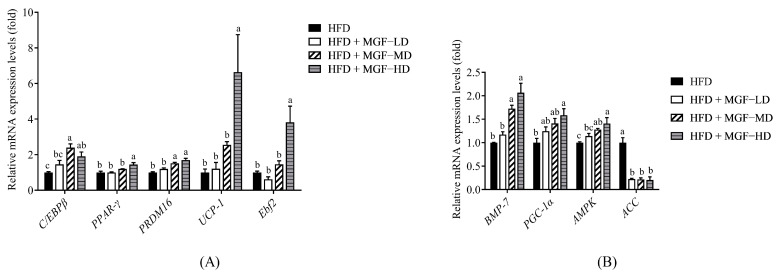
Effect of the Mei-Gin formula on gene expression of (**A**) thermogenesis and (**B**) mitochondrial biogenesis and synthesis in BAT of obese rats induced by a high-fat diet. Data are presented as the mean ± SEM. Different letters for each test parameter are considered for statistical significance (*p* < 0.05). *C/EBPβ*, CCAAT/enhancer-binding protein beta; *PPAR-*γ, peroxisome proliferator-activated receptor-gamma; *PRDM16*, PR domain containing 16; *UCP-1*, uncoupling protein 1; *Ebf2*, EBF transcription factor 2; *BMP-7*, bone morphogenetic protein 7; *PGC-1*α, peroxisome proliferator-activated-gamma coactivator 1 alpha; *ACC*, acetyl-CoA carboxylase.

**Figure 6 foods-12-03539-f006:**
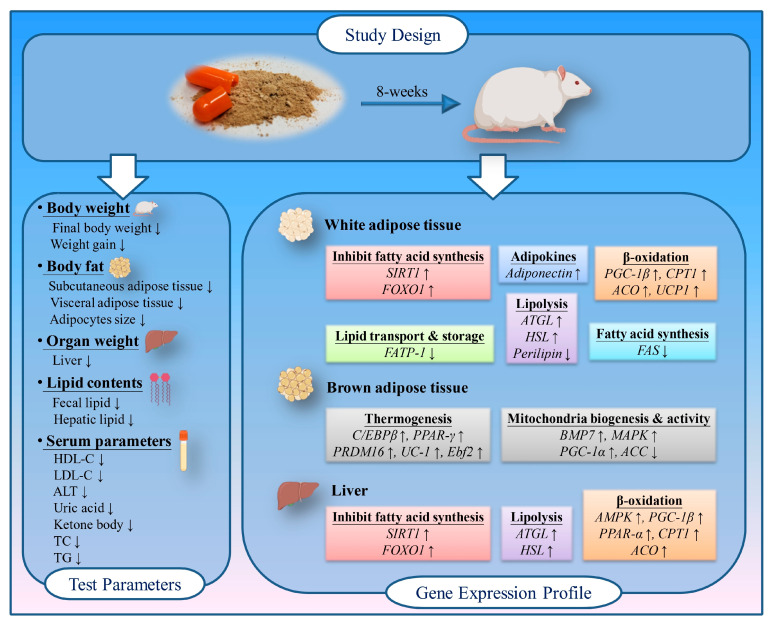
The schematic abstract of the anti-obesity effects of MGF in HFD-induced obesity rats. MGF exerted anti-obesity actions through suppressing body weight and fat accumulation and liver and serum lipid content; with regard to its mechanistic pathways, these involve the downregulation of fatty acid synthesis, lipid transport, and storage, and the upregulation of adipokines, β-oxidation, lipolysis, thermogenesis, mitochondria biogenesis, and related genes’ expression. Created with BioRender.com.

**Table 1 foods-12-03539-t001:** Diet formulation.

Ingredients	AIN-93G		AIN-93G-Based HFD
		g/kg diet	
corn starch	397.500		147.500
casein	200.000		200.000
maltodextrin	132.000		132.000
powdered cellulose	50.000		50.000
mineral mix (AIN-93G MX)	35.000		35.000
vitamin mix (AIN-93G VX)	10.000		10.000
L-cystine	3.000		3.000
choline bitartrate	2.500		2.500
t-butyldroquinone	0.014		0.014
sucrose	0.000		100.000
soybean oil	0.000		70.000
lard	0.000		250.000

**Table 2 foods-12-03539-t002:** Effect of the Mei-Gin formula on metabolic parameters and organ weights in high-fat diet-induced obese rats.

Group	ND	HFD	HFD + MGF−LD	HFD + MGF−MD	HFD + MGF−HD
Initial body weight (g)	231.66 ± 3.21 ^a^	232.66 ± 3.42 ^a^	232.61 ± 3.25 ^a^	231.14 ± 3.23 ^a^	231.18 ± 3.24 ^a^
Final body weight (g)	523.29 ± 9.17 ^b^	594.77 ± 18.11 ^a^	549.23 ± 11.73 ^b^	540.41 ± 15.24 ^b^	539.83 ± 13.92 ^b^
Weight change (g)	291.63 ± 10.67 ^b^	362.11 ± 17.03 ^a^	316.63 ± 11.29 ^b^	309.27 ± 13.83 ^b^	308.66 ± 13.77 ^b^
Feed intake (g/rat/day)	84.25 ± 3.18 ^ab^	83.33 ± 4.60 ^ab^	69.83 ± 4.32 ^b^	76.83 ± 4.30 ^ab^	89.33 ± 10.68 ^a^
Energy intake (kcal/rat/day)	43.00 ± 2.80 ^ab^	46.67 ± 3.05 ^a^	38.67 ± 2.16 ^bc^	36.00 ± 2.36 ^bc^	34.17 ± 2.50 ^c^
Feed efficiency (%)	0.56 ± 0.01 ^a^	0.56 ± 0.02 ^a^	0.50 ± 0.01 ^b^	0.53 ± 0.01 ^ab^	0.57 ± 0.01 ^a^
Water intake (mL/rat/day)	7.03 ± 0.05 ^a^	6.84 ± 0.16 ^a^	6.32 ± 0.07 ^b^	5.19 ± 0.05 ^c^	6.31 ± 0.06 ^b^
Heart (g)	1.53 ± 0.04 ^a^	1.64 ± 0.02 ^a^	1.65 ± 0.08 ^a^	1.55 ± 0.04 ^a^	1.57 ± 0.04 ^a^
Liver (g)	16.92 ± 0.51 ^a^	18.41 ± 0.69 ^a^	15.01 ± 0.40 ^b^	15.00 ± 0.53 ^b^	13.93 ± 0.49 ^b^
Spleen (g)	1.00 ± 0.04 ^a^	0.99 ± 0.03 ^a^	0.90 ± 0.04 ^a^	0.91 ± 0.02 ^a^	0.94 ± 0.04 ^a^
Lung (g)	2.00 ± 0.06 ^a^	2.17 ± 0.07 ^a^	2.27 ± 0.16 ^a^	2.25 ± 0.09 ^a^	2.17 ± 0.09 ^a^
Kidney (g)	3.64 ± 0.09 ^a^	3.75 ± 0.06 ^a^	3.55 ± 0.07 ^a^	3.58 ± 0.04 ^a^	3.59 ± 0.04 ^a^

Data are presented as the mean ± SEM. Different letters for each test parameter are considered for statistical significance (*p* < 0.05). ND, normal diet; HFD, high-fat diet; MGF, Mei-Gin formula. Weight change (g) = final body weight (g) − initial body weight (g).

**Table 3 foods-12-03539-t003:** Effect of the Mei-Gin formula on adipose tissue in high-fat diet-induced obese rats.

Weight (mg/g Rat)	ND	HFD	HFD + MGF−LD	HFD + MGF−MD	HFD + MGF−HD
Total body fat	107.98 ± 8.21 ^b^	151.32 ± 11.06 ^a^	121.50 ± 7.14 ^b^	121.96 ± 11.47 ^b^	116.21 ± 5.21 ^b^
Subcutaneous adipose tissue	35.34 ± 3.86 ^b^	50.97 ± 4.54 ^a^	36.64 ± 2.17 ^b^	39.63 ± 5.24 ^b^	38.27 ± 1.72 ^b^
retroperitoneal adipose	22.20 ± 2.96 ^a^	30.63 ± 3.16 ^a^	22.57 ± 1.74 ^a^	26.51 ± 3.95 ^a^	23.57 ± 1.15 ^a^
inguinal adipose	13.13 ± 1.34 ^b^	20.34 ± 2.01 ^a^	14.07 ± 0.83 ^b^	13.12 ± 1.49 ^b^	14.69 ± 0.74 ^b^
Visceral adipose tissue	72.65 ± 4.87 ^b^	100.35 ± 6.84 ^a^	84.86 ± 5.99 ^ab^	82.33 ± 6.59 ^b^	77.94 ± 3.78 ^b^
perirenal adipose	30.13 ± 2.56 ^b^	41.19 ± 2.91 ^a^	36.00 ± 2.15 ^ab^	33.88 ± 3.65 ^ab^	32.12 ± 1.51 ^b^
epididymal adipose	24.15 ± 1.73 ^b^	31.95 ± 2.30 ^a^	27.27 ± 2.29 ^ab^	26.26 ± 1.71 ^ab^	26.40 ± 1.26 ^ab^
mesenteric adipose	18.37 ± 1.24 ^b^	27.21 ± 2.48 ^a^	21.59 ± 2.04 ^ab^	22.19 ± 1.99 ^ab^	19.42 ± 1.82 ^b^
Brown adipose tissue	4.09 ± 0.42 ^a^	5.04 ± 0.62 ^a^	4.52 ± 0.38 ^a^	4.95 ± 0.49 ^a^	4.32 ± 0.53 ^a^

Data are presented as the mean ± SEM. Different letters for each test parameter are considered for statistical significance (*p* < 0.05).

**Table 4 foods-12-03539-t004:** Effect of the Mei-Gin formula on serum biochemical parameters in obese rats induced by a high-fat diet.

Group	ND	HFD	HFD + MGF−LD	HFD + MGF−MD	HFD + MGF−HD
Glucose (mg/dL)	210.46 ± 11.09 ^a^	227.64 ± 12.74 ^a^	241.66 ± 10.72 ^a^	210.03 ± 9.35 ^a^	210.79 ± 9.19 ^a^
TG (mg/dL)	107.91 ± 7.55 ^a^	84.07 ± 8.00 ^b^	82.19 ± 5.49 ^bc^	76.56 ± 3.41 ^bc^	65.62 ± 2.12 ^c^
Total cholesterol (mg/dL)	125.41 ± 4.80 ^a^	107.10 ± 4.14 ^b^	98.98 ± 3.34 ^bc^	91.57 ± 3.88 ^c^	90.17 ± 3.70 ^c^
HDL-C (mg/dL)	59.33 ± 1.34 ^a^	59.33 ± 1.14 ^a^	54.08 ± 1.20 ^b^	49.08 ± 1.23 ^c^	45.58 ± 0.70 ^d^
LDL-C (mg/dL)	35.42 ± 1.14 ^a^	32.58 ± 0.93 ^ab^	30.17 ± 1.13 ^bc^	27.50 ± 1.23 ^c^	28.25 ± 0.63 ^c^
LDL-C/HDL-C ratio	0.60 ± 0.02 ^ab^	0.55 ± 0.01 ^b^	0.56 ± 0.01 ^b^	0.56 ± 0.03 ^b^	0.62 ± 0.04 ^a^
AST (U/L)	84.25 ± 3.18 ^ab^	83.33 ± 4.60 ^ab^	69.83 ± 4.32 ^b^	76.83 ± 4.30 ^ab^	89.33 ± 10.68 ^a^
ALT (U/L)	43.00 ± 2.80 ^ab^	46.67 ± 3.05 ^a^	38.67 ± 2.16 ^bc^	36.00 ± 2.36 ^bc^	34.17 ± 2.50 ^c^
Creatinine (mg/dL)	0.56 ± 0.01 ^a^	0.56 ± 0.02 ^a^	0.50 ± 0.01 ^b^	0.53 ± 0.01 ^ab^	0.57 ± 0.01 ^a^
Uric acid (mg/dL)	7.03 ± 0.05 ^a^	6.84 ± 0.16 ^a^	6.32 ± 0.07 ^b^	5.19 ± 0.05 ^c^	6.31 ± 0.06 ^b^
Ketone body (mg/dL)	6.10 ± 0.17 ^b^	7.27 ± 0.10 ^a^	6.07 ± 0.18 ^b^	5.28 ± 0.16 ^c^	5.68 ± 0.12 ^bc^
Na^+^ (mmol/L)	146.92 ± 0.29 ^a^	146.50 ± 0.31 ^a^	146.42 ± 0.29 ^a^	146.00 ± 0.33 ^a^	146.25 ± 0.28 ^a^
K^+^ (mmol/L)	4.54 ± 0.02 ^a^	4.58 ± 0.02 ^a^	4.53 ± 0.02 ^a^	4.56 ± 0.01 ^a^	4.58 ± 0.03 ^a^
Cl^−^ (mmol/L)	105.83 ± 0.21 ^a^	106.33 ± 0.36 ^a^	106.00 ± 0.28 ^a^	105.67 ± 0.22 ^a^	105.92 ± 0.26 ^a^

Data are presented as the mean ± SEM. Different letters for each test parameter are considered for statistical significance (*p* < 0.05). HDL-C, high-density lipoprotein cholesterol; LDL-C, low-density lipoprotein cholesterol; AST, aspartate aminotransferase; ALT, alanine aminotransferase.

## Data Availability

Not applicable.

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
