# Peer review of "Mei-Gin Formula Ameliorates Obesity through Lipolysis, Fatty Oxidation, and Thermogenesis in High-Fat Diet-Induced Obese Rats"

_foods, 2023, doi:10.3390/foods12193539_

Round 1

Reviewer 1 Report

This paper explores the biochemical changes that ocurr after the administration of three different doses of a new natural anti-obesity formula in obese rats. Looking at the literature, this is the second work on MGF of the group. In this study they aim to explain the pharmacological effects of the formula through histopathological, biochemical and expression changes that ocurr after 8 weeks administration of three different formula doses.

My main concern resides in the lack of Normal Diet controls for each Mei-Gin dose. The lack of these controls makes it difficult to define the precise effect of the MGF, above all at the biochemical level. Considering the HFD individuals did not exert any metabolic alteration in the plasma yet, but had more weight and fat in their bodies, we need to know which basal effects are observed in normal individual to then precisely determine the effects on fat individuals.

Another very important missing point is the description of all gene expression in the ND group. This needs to be included.

After including this, I would recommend to improve the discussion

Discussion in lines 380-386 should include, what it means that HFD did not show bichemical serum changes even if they were "fat", and how this affects the interpretation of the MGF effects and what should be done to improve the conclusions of the effects.

Also in the discussion about the gene expression I highly suggest to perform a diagram in which the whole differential expression pattern is explained by tissue and by metabolic pathway. 

Perilipin expression results are missing in the discussion and should be included.

And once again, it should be carefully explained, how these gene expression differences impact on metabolism, considering, there where normal serum lipid concentrations. 

Finally,  the phrase (lines 459-462):" Furthermore, these in vivo evidences indicated that lipolysis, fatty oxidation, and thermogenesis generated central mechanistic pathways through MGF exhibit a strong effect against HDF-induced obesity in rats" is inappropriate due to my previous comments: we lack the ND controls, and you must consider the ND expression pattern in order to improve conclusions about the metabolic and biochemical effects of MGF.

Author Response

Comments and Suggestions for Authors

Reviewer #1:

This paper explores the biochemical changes that ocurr after the administration of three different doses of a new natural anti-obesity formula in obese rats. Looking at the literature, this is the second work on MGF of the group. In this study they aim to explain the pharmacological effects of the formula through histopathological, biochemical and expression changes that ocurr after 8 weeks administration of three different formula doses.

  1. My main concern resides in the lack of Normal Diet controls for each Mei-Gin dose. The lack of these controls makes it difficult to define the precise effect of the MGF, above all at the biochemical level. Considering the HFD individuals did not exert any metabolic alteration in the plasma yet, but had more weight and fat in their bodies, we need to know which basal effects are observed in normal individual to then precisely determine the effects on fat individuals.

Reply: Thanks for your commons and suggestions. We greatly appreciated receiving your responses that help us strengthening the paper work. It’s a very good opinion that extra ND controls for each Mei-Gin formula intervention groups would helping us to better clarify the biological effects of Mei-Gin formula. Previously, our preliminary data show that MGF-7 effectively reduced high-fat diet caused body weight, liver weight, and total body fat gain in rats and exerted the most therapeutic potential for obesity. Therefore, the present work focus on evaluating the anti-obesity effects and underlying mechanisms of MGF-7 at different dose based on our previous finding. Indeed, we did not explore the biological effects of MGF on normal individuals (normal diet). In general, such experimental design using diet-induced obesity animal was a well established model and long-term used (Jiao etal., 2019; Ding et al., 2019; Suk et al., 2017), on the basis of verifying the anti-obesity ability of agent/ compound. We thought it’s an interesting issue/work that worthy to explore in the future, in order to better characterize the bioactivity of MGF. We thus state the study limitation in the discussion section. P13, Line 502-507.

Besides, we speculated that HFD feeding did not increased serum lipid concentration in rats might due to low-carbohydrate dietary effects. We have state the dietary effect on serum parameters in discussion section. P12, Line 420-427.

Ref:

  1. Blueberry polyphenols extract as a potential prebiotic with anti-obesity effects on C57BL/6 J mice by modulating the gut microbiota. Xinyao Jiao, Yuehua Wang, Yang Lin, Yuxi Lang, Enhui Li, Xiuyan Zhang, Qi Zhang, Ying Feng, Xianjun Meng, Bin Li. J Nutr Biochem. 2019 Feb; 64:88-100. doi: 10.1016/j.jnutbio.2018.07.008.
  2. Anti-Obesity Effect of Diphlorethohydroxycarmalol Isolated from Brown Alga Ishige okamurae in High-Fat Diet-Induced Obese Mice. Yuling Ding, Lei Wang, SeungTae Im, Ouibo Hwang, Hyun-Soo Kim, Min-Cheol Kang, Seung-Hong Lee. Mar Drugs. 2019 Nov 10;17 (11): doi: 10.3390/md17110637.
  3. Gingerenone A, a polyphenol present in ginger, suppresses obesity and adipose tissue inflammation in high-fat diet-fed mice. Sujin Suk, Gyoo Taik Kwon, Eunjung Lee, Woo Jung Jang, Hee Yang, Jong Hun Kim, N R Thimmegowda, Min-Yu Chung, Jung Yeon Kwon, Seunghee Yang, Jason K Kim, Jung Han Yoon Park, Ki Won Lee.Mol Nutr Food Res. 2017 Oct;61(10):10.1002/mnfr.201700139. doi: 10.1002/mnfr.201700139.
  4. Another very important missing point is the description of all gene expression in the ND group. This needs to be included. After including this, I would recommend to improve the discussion

Reply: Thanks for your commons and suggestions. The HFD feeding rats significantly reduced genes expression involved in liver fatty acid synthesis and increased adipose β-oxidation related genes expression have been previously confirmed. Other study also only compared protein expression of lipid metabolism i.e. PPARγ, C/EBPα, SREBP-1c, FABP4, and FAS between HFD and intervention groups (ref). Therefore, we did not explore the basic expression of lipid metabolism related genes in ND group. We have briefly stated in the discussion section. P13, Line 459-462.

Ref:

  1. Anti-Obesity Effect of Diphlorethohydroxycarmalol Isolated from Brown Alga Ishige okamurae in High-Fat Diet-Induced Obese Mice. Yuling Ding, Lei Wang, SeungTae Im, Ouibo Hwang, Hyun-Soo Kim, Min-Cheol Kang, Seung-Hong Lee. Mar Drugs. 2019 Nov 10;17(11):637.  doi: 10.3390/md17110637.
  2. Discussion in lines 380-386 should include, what it means that HFD did not show bichemical serum changes even if they were "fat", and how this affects the interpretation of the MGF effects and what should be done to improve the conclusions of the effects.

Reply: Thanks for your commons and suggestions. According to the reviewer’s suggestion, we have state the dietary effect on serum parameters and the anti-obesity actions of MGF. P12, Line 420-427.  

  1. Also in the discussion about the gene expression I highly suggest to perform a diagram in which the whole differential expression pattern is explained by tissue and by metabolic pathway. 

Reply: Thanks for your commons and suggestions. According to the reviewer’s suggestion, we have added a graphical figure to summarize the anti-obesity effects of longan and better understanding the differential in genes expression among organ or tissue. P14, Line 518-538 (figure 6).

  1. Perilipin expression results are missing in the discussion and should be included.

Reply: Thanks for your commons and suggestions. According to the reviewer’s suggestion, we have state the result of Perilipin mRNA expression in the discussion section. P13, Line 464-465.

  1. And once again, it should be carefully explained, how these gene expression differences impact on metabolism, considering, there where normal serum lipid concentrations. 

Reply: Thanks for your commons and suggestions. Unfortunately, normal serum lipid levels are very from studies in HFD-induced rat model (Ting et al., 2018, Xu et al., 2015, Khalaf et al., 2023). And according to the reviewer’s suggestion, we have added a graphical figure to summarize the anti-obesity effects of MGF and better understanding the impact of differential in genes expression on lipid metabolism. P14, Line 518-538 (figure 6).

Ref:

  1. Antiobesity Efficacy of Quercetin-Rich Supplement on Diet-Induced Obese Rats: Effects on Body Composition, Serum Lipid Profile, and Gene Expression. Yuwen Ting, Wei-Tang Chang, Duen-Kai Shiau, Pei-Hsuan Chou, Mei-Fang Wu, Chin-Lin Hsu. J Agric Food Chem. 2018 Jan 10;66 (1):70-80. doi: 10.1021/acs.jafc.7b03551.
  2. The anti-obesity effect of green tea polysaccharides, polyphenols and caffeine in rats fed with a high-fat diet. Yan Xu1, Min Zhang, Tao Wu, ShengDong Dai, Jinling Xu, Zhongkai Zhou. Food Funct. 2015 Jan;6 (1):297-304. doi: 10.1039/c4fo00970c.
  3. Acacia nilotica stem bark extract ameliorates obesity, hyperlipidemia, and insulin resistance in a rat model of high fat diet-induced obesity. Samar S Khalaf, Ola A Shalaby, Ahmed R Hassan, Mohamed K El-Kherbetawy, Eman T Mehanna. J Tradit Complement Med. 2023 Mar 8;13 (4):397-407.  doi: 10.1016/j.jtcme.2023.03.005.
  4. Finally, the phrase (lines 459-462):" Furthermore, these in vivo evidences indicated that lipolysis, fatty oxidation, and thermogenesis generated central mechanistic pathways through MGF exhibit a strong effect against HDF-induced obesity in rats" is inappropriate due to my previous comments: we lack the ND controls, and you must consider the ND expression pattern in order to improve conclusions about the metabolic and biochemical effects of MGF.

Reply: Thanks for your commons and suggestions. We have state the study limitation in the discussion section. P13, Line 502-507.

Reviewer 2 Report

I commend your work on the article entitled "Mei-Gin formula ameliorates obesity through lipolysis, fatty oxidation, and thermogenesis in high-fat diet-induced obese rats". It addresses a highly intriguing topic with a wealth of knowledge and information. I wholeheartedly recommend that this paper be the major revision. This is indeed a novel and captivating subject that I have not encountered before. Nevertheless, I would like to offer some suggestions to further enhance the article:

- In the abstract, the results need to be rewritten, mention several parameters that indicate significance values.

- Line76-78, authors mention several combinations of plant extracts to create the Mei-Gin formula; Please provide the Latin name of each plant/species used.

- Please provide a section on methods, how the Mei-Gin formula is made and mention the compounds contained in it that were observed by the Authors in previous publications.

- Please register this research protocol on the website: https://preclinicaltrials.eu; it can be accessed anytime and for free. If it has been registered, then provide the registration number and call it on the method.

- Provide details of the pellet diet given to rats, both normal and HFD, which must be clear with detailed composition.

- Clarify how many animals/rats are used, and how many per group? What formula is used to select the number of rats? it should be mentioned in the method.

- What motivates the authors to use the selected dose of the Mei-Gin formula? state clearly the background to the selection of the dose!

- Discuss and compare the results of this study with other studies on the same topic: https://doi.org/10.3390/nu15040909; https://doi.org/10.3390/antiox12081555.

I believe that implementing these suggestions will significantly enhance the overall quality and impact of your article.

Author Response

Reviewer #2:

I commend your work on the article entitled "Mei-Gin formula ameliorates obesity through lipolysis, fatty oxidation, and thermogenesis in high-fat diet-induced obese rats". It addresses a highly intriguing topic with a wealth of knowledge and information. I wholeheartedly recommend that this paper be the major revision. This is indeed a novel and captivating subject that I have not encountered before. Nevertheless, I would like to offer some suggestions to further enhance the article:

  1. - In the abstract, the results need to be rewritten, mention several parameters that indicate significance values.

Reply: Thanks for your commons and suggestions. According to the reviewer’s suggestion, we have rewrite experimental results in abstract, state metabolic parameters in detail. P1, Line 28-32.

  1. - Line76-78, authors mention several combinations of plant extracts to create the Mei-Gin formula; Please provide the Latin name of each plant/species used.

Reply: Thanks for your commons and suggestions. According to the reviewer’s suggestion, the Latin name has been added in parentheses after each herbal plant in the text. P2, Line 61-62, 72.

  1. - Please provide a section on methods, how the Mei-Gin formula is made and mention the compounds contained in it that were observed by the Authors in previous publications.

Reply: Thanks for your commons and suggestions. According to the reviewer’s suggestion, we have briefly state the preparation of MGF-7 in method section. P3, Line 108-115.

  1. - Please register this research protocol on the website: https://preclinicaltrials.eu; it can be accessed anytime and for free. If it has been registered, then provide the registration number and call it on the method.

Reply: Thanks for your commons and suggestions. According to the reviewer’s suggestion, we have provided research information on PCT (ID: PCTE0000404). P3, Line 137-138.

  1. - Provide details of the pellet diet given to rats, both normal and HFD, which must be clear with detailed composition.

Reply: Thanks for your commons and suggestions. According to the reviewer’s suggestion, we have provided the diet formulation in material and method section. P3, Line 139-P4, Line140.

  1. - Clarify how many animals/rats are used, and how many per group? What formula is used to select the number of rats? it should be mentioned in the method.

Reply: Thanks for your commons and suggestions. According to the reviewer’s suggestion, a total of sixty rats were used in the present study and twelve for each group. P3, Line 117, 126. 

  1. - What motivates the authors to use the selected dose of the Mei-Gin formula? state clearly the background to the selection of the dose!

Reply: Thanks for your commons and suggestions. According to the reviewer’s suggestion, this present study evaluates the efficacies of MGF in the HFD induced rat model and aims on verifying the therapeutic dosage with low and/ or safer concentration. The motivation and dosage selected were state in the text. P2, Line 93-95 and P3, Line 120-25.

  1. - Discuss and compare the results of this study with other studies on the same topic: https://doi.org/10.3390/nu15040909; https://doi.org/10.3390/antiox12081555.

Reply: Thanks for your commons and suggestions. According to the reviewer’s suggestion, the specific study about anti-obesity dietary supplements has been discussed in discussion section. P12, Line 413-418. Ref 37, 38

I believe that implementing these suggestions will significantly enhance the overall quality and impact of your article.

Round 2

Reviewer 1 Report

I liked very much the answers and above all, the diagram of figure 6.

nice job

Reviewer 2 Report

Authors revised the manuscript according to reviewers comments.